# SNAP timing and food insecurity

**Christian A. Gregory** [1]*, **Jessica E. Todd** [2]

1 Food Economics Economics Division, Economic Research Service, Food Assistance Branch, U.S. Department of Agriculture, Kansas City, Missouri, United States of America, 2 Resource and Rural Economics Division, Economic Research Service, Farm Economy Branch, U.S. Department of Agriculture, Washington, District of Columbia, United States of America

* Christian.gregory@usda.gov

## Abstract

This paper makes several contributions to the literature regarding the measurement of food insecurity and implications for estimating factors that affect this outcome. First, we show that receipt of benefits from the Supplemental Nutrition Assistance Program (SNAP) has a systematic effect on responses to questions in the 12-month food security module (FSM). We find that the probability of affirming more severe food hardships items, and the probability of being classified as having very low food security (VLFS), is higher just before and just after households receive their benefits. This leads to an under-estimate of VLFS by 3.2 percentage points for the SNAP sample (about 17 percent of prevalence). We also provide informative bounds on the relationship between SNAP and VLFS and show that the treatment effect of SNAP on VLFS is also likely underestimated.

## Introduction

Officially defined, food security means having access at all times to enough food for an active, healthy life. Since 1995, the United States' food security statistics have been produced using data from the Food Security Module (FSM), which asks respondents whether or not a number of food-related conditions were experienced in their households, ranging from worrying about running out of food to having to skip meals and even having lost weight because there wasn't enough money in the household for food. In 2019, USDA reported that 13.7 million households were food insecure, and 5.6 million experienced very low food security, meaning that at least one household member altered their eating patterns and reduced food intake because they lacked sufficient resources for food. (The main FSM is collected from a sample of households without children interviewed during a 10-day period in December of each year (Coleman-Jensen et al., 2019). An expanded 18-item module captures details about children's experiences and is asked of households with children. Households affirming 0 to 2 conditions are classified as food secure. Households affirming more than 2 conditions (food insecure households) are classified as having low or very low food security. For households without children, low food security means that respondents have affirmed 3–5 conditions in the module (3–7 conditions for households with children); very low food security implies affirmations of 6 or more items (8 or more items in households with children).)

The Supplemental Nutrition Assistance Program (SNAP, formerly the Food Stamp Program), is the largest food assistance program in the United States and aims to reduce food

**Data Availability Statement:** Data are from the National Health and Nutrition Examination Survey, a federal data collection. https://www.cdc.gov/nchs/nhanes/index.htm.

**Competing interests:** The authors have declared that no competing interests exists.

insecurity and improve nutrition among low-income households. In 2019, it provided an average of $258 in monthly benefits to 18 million households [1]. A large and growing literature documents a variety of patterns of behaviors and outcomes (mainly declines in expenditures and dietary intake at the end of the benefit month) associated with monthly receipt of SNAP benefits–the SNAP cycle [2–7], which may be in part related to the fact that SNAP benefits may not be sufficient to meet participants' food needs [6–11]. Additional work has shown that adults bear the brunt of cyclical food insecurity, with school nutrition programs bolstering the intakes of children [12]. The monthly SNAP distribution cycle has also been shown to affect criminal activity [13] and emergency room visits [14].

This paper extends research about the SNAP cycle–the behaviors associated with the monthly distribution of SNAP benefits–by investigating how the receipt of SNAP benefits affects responses to the questions in the 12-month food security module (FSM). The importance of this extension is several-fold. First, while prior research on the SNAP cycle has tended to concentrate on spending or consumption outcomes, we focus on the psychological effect of SNAP receipt on responses that affect our understanding of food insecurity prevalence. Second, while prior research has established these effects with respect to the 30-day FSM, we study the 12-month FSM, which is used to obtain official statistics on food insecurity in the U. S. [15]; this examination brings us closer to understanding potential biases in statistics derived from that measure. Third, this investigation offers a glimpse into the response patterns to items whose reference period (12-month FSM) is different from the event that frames or makes it salient (30-day SNAP receipt).

In particular, we look at whether and how the recent receipt of SNAP benefits affects the responses to the 12-month FSM questions in the National Health and Nutrition Examination Survey (NHANES). We find that the pattern of responses to the FSM in part follows the SNAP consumption cycle, with affirmations of food-insecure conditions more likely at the end of the benefit month. However, we also find that affirmations of some items also increase in the first week or ten days of the SNAP month, consistent with [16]. In examining this feature of the response patterns to the FSM questions, we highlight two aspects of these responses. First, although the responses are to the *12-month* FSM module questions in NHANES, they are highly correlated with the *monthly* SNAP distribution cycle. Second, the items that vary the most relative to SNAP receipt are the *more severe* adult food security items and the summary measure estimating the probability of very low food security (VLFS) among adults. This differs from Gregory and Smith (2019), who found higher probabilities for the *least severe* items in the 30-day FSM. We suspect that this has to do with the way that respondents summarize the previous year's experiences of food hardships. As the work of Kahneman and Krueger [17] and others has shown, remembered utility is different from experienced utility; a summary response of a year's worth of experience is likely to down weight less recent experiences–positive or negative–and up weight the average of the most recent period's peak or trough experiences. The fact that more severe items appear affected by SNAP timing is consistent with this accounting for remembered utility.

Although we cannot test for remembered utility directly, we investigate the issue indirectly using a number of approaches. First, we compare the pattern of responses to other questions in the NHANES survey that might be likely to exhibit patterns similar to those of the 12-month FSM. For example, we examine reported household food spending over the last 30 days, which we might expect to be correlated with SNAP receipt. We additionally test questions with 12-month reference period about frequency and intensity of alcohol consumption. None of the responses to these questions exhibits any correlation with when a household receives their SNAP benefits. Finally, we also examine responses to the 30-day *individual* food security questions, which were only asked of a subset of our full sample during a portion of the

period covered by our data. These responses exhibit patterns consistent with recall bias correlated with the SNAP cycle, with increases in the probability of affirmating food insecure conditions at the end of the month, but not at the beginning. This contrast suggests that SNAP receipt brings to mind more severe experiences of food insecurity from the previous 12 months, which prompts affirmation of those items.

There are several important implications of these findings. First, in the presence of non-uniform distribution of SNAP benefit delivery but uniform survey administration over the month, estimates of VLFS prevalence will be under-estimated. The extent of this bias for population estimates obviously depends upon SNAP participation, but if responses in the "middle" of the SNAP month under-report 12-month food hardships, underestimates of VLFS are on the order of 3.2 percentage points for SNAP participants, about 17 percent of prevalence for this group. For all low-income households (SNAP participants and nonparticipants), the underestimate is about 1.2 percentage points, or about 9 percent of prevalence. In other words, our results suggest that low-income households, and especially SNAP households, experience more food insecure conditions, including disruptions in intake, than are currently measured. A second implication is that estimates of the effect of SNAP on reducing VLFS is likely to be biased downward. We show a wide range of possible upper bounds on SNAP's effect on VLFS. All of them indicate substantial differences in average treatment effects depending on the time since last SNAP receipt.

The remainder of this paper is organized as follows. In the next section, we describe the data that we use, and the next section discusses methods. The last two sections show results and discuss their implications, respectively.

## Data

The data come from the National Health and Nutrition Examination Survey (NHANES), which is a probability-weighted, nationally representative data collection with a complex survey design. NHANES is collected continuously, but data are released in two-year waves and includes over-samples of Hispanic, non-Hispanic black, low-income white and other non-Hispanic white persons over 80 years old. NHANES is unique among federal data collections in that it includes a series of in-person and over-the-phone interviews as well as medical examinations in a mobile examination center (MEC). We use four waves of the survey, covering 2007 to 2014.

### Measuring food security

The 12-month food security module (FSM) is a series of 10 (18) questions for households without (with) children under age 18. The first 10 items ask about the household as a whole and about behaviors or situations among adults that are known to be correlated with lack of access to adequate food. The final 8 items ask about children's experiences with food hardship among households that include children. In NHANES, the FSM is administered during the household interview, which is the first data collection instrument for the vast majority of surveyed households. SNAP participation and last receipt date are also obtained during this interview.

We concentrate on the adult food security module questions, since they provide a common base of comparison across all households. The 10 items in the FSM are listed in Box 1. In keeping with current guidance, for questions that ask whether a condition occurred often, sometimes, or never, we count any response but never as an affirmation [18]. For questions that ask whether conditions occurring almost every month, some months . . ., or in only 1 or 2 months, we count any response other than 1 or 2 months as affirmations. We classify households according to the number of affirmations to all of the questions: households with no affirmations have high food security; those with one or two affirmations are counted as marginally food secure; those with three to five affirmations have low food security, and households with

> ### Box 1. Questions used to assess the food security of households in the NHANES food security survey
>
> 1. "We worried whether our food would run out before we got money to buy more." Was that often, sometimes, or never true for you in the last 12 months?
>
> 2. "The food that we bought just didn't last and we didn't have money to get more." Was that often, sometimes, or never true for you in the last 12 months?
>
> 3. "We couldn't afford to eat balanced meals." Was that often, sometimes, or never true for you in the last 12 months?
>
> 4. In the last 12 months, did you or other adults in the household ever cut the size of your meals or skip meals because there wasn't enough money for food? (Yes/No)
>
> 5. (If yes to question 4) How often did this happen—almost every month, some months but not every month, or in only 1 or 2 months?
>
> 6. In the last 12 months, did you ever eat less than you felt you should because there wasn't enough money for food? (Yes/No)
>
> 7. In the last 12 months, were you ever hungry, but didn't eat, because there wasn't enough money for food? (Yes/No)
>
> 8. In the last 12 months, did you lose weight because there wasn't enough money for food? (Yes/No)
>
> 9. In the last 12 months did you or other adults in your household ever not eat for a whole day because there wasn't enough money for food? (Yes/No)
>
> 10. (If yes to question 9) How often did this happen—almost every month, some months but not every month, or in only 1 or 2 months?
>
> End of Box 1

more than 5 affirmations have very low food security [18]. Food insecure households are those who have low or very low food security.

## Identifying SNAP participants

NHANES contains a series of questions establishing SNAP participation. The sample respondent is asked whether the household has ever received SNAP benefits, whether it has received benefits in the last 12 months, whether the household currently receives SNAP, and when SNAP was last received. In public-use data, the length of time (in days) since the last receipt is provided, as calculated from the interview date and the reported last-receipt-of-SNAP date. We count any household that has received SNAP in the last 30 days as a SNAP participant.

## Other variables of interest

NHANES contains a rich set of demographic and health variables that we use to reduce the variance of our predictions in our regression models. Because food security is a household phenomenon, we focus on household level characteristics that might be related to food security status: we include gender, age, race, and educational status of the survey respondent, education

of the household reference person (which may not be the survey respondent), whether the household has at least $2000 in savings, and household income relative to the federal poverty level for household size. We show summary statistics for these characteristics below.

## Methods

We employ several estimation and prediction strategies in this paper. First, we use non-parametric regressions of days since SNAP receipt on each of the food security outcomes and indicators for food insecurity and very low food security to understand how closely affirmations to these questions are correlated with receipt of SNAP benefits. Following Gregory and Smith [16], we call the time closest to receipt of benefits the salience window, and expect that this is when the FSM questions are more salient to SNAP recipient respondents. These non-parametric regressions serve as *prima facie* evidence of variation in affirmation of the FSM questions over the SNAP month.

Second, we estimate regressions that condition on household observable characteristics (defined above) and indicators for whether the household answered the FSM questions within the salience window. We then estimate the marginal effect of being in the salience window of affirming the FSM items, food insecurity, or very low food security.

We use these regression results to predict the potential underestimate of very low food security due to the timing of SNAP receipt. In particular, we draw from the distribution of marginal effects estimated for each item in the FSM and then, using the bootstrap method for complex surveys described by Kolenikov [19] we reassign affirmations to each of the food security questions. (The bootstrap procedure samples randomly from the number of primary sampling units (psu) or clusters within a given stata. The number of clusters to sample from each strata is the source of differences in the characteristics of bootstrap estimators. We choose $m_h = n_h - 1$ clusters to sample in each strata, where $n_h$ is the total number of psus per strata. Both Rao and Wu [20] and Kolenikov [19] suggest this as a good choice that provides stability in the estimates and doesn't require rescaling of observed variables.) We sum the simulated affirmations and calculate the simulated incidence of very low food insecurity for SNAP recipients, households with incomes below 200% of the federal poverty line, and for the whole population. We compare these to the survey-estimated unconditional prevalence for each of these groups.

Finally, we use this simulated prevalence to examine differences in the upper bound on the treatment effect of SNAP on very low food security for those households within and outside the salience window. A principle difficulty here is the absence of a credible identification strategy for point-identifying causal effects. However, the non-parametric bounding literature partially identifies the treatment effect of SNAP on food insecurity by using elementary laws of probability and prior information about the selection and treatment effects. Two prominent assumptions in this literature are monotone treatment response (MTR)–meaning that SNAP receipt cannot *harm* a household in terms of food security–and monotone treatment selection (MTS)–in this case that SNAP participants are more likely to experience very low food security with or without SNAP [21, 22]. We show that if we assume MTR and MTS we can get informative bounds on the potential decreases in very low food security due to SNAP participation. Because program participation is mis-measured in survey data — mostly, under-reported [23–26]—we also estimate how these bounds vary with different amounts of assumed under-reporting in SNAP participation.

## Results

### Non-parametric regressions and descriptive statistics

Fig 1 shows the results of non-parametric regressions of days-since-SNAP on each of the food security module questions as well as the binary variables indicating food insecurity and very-

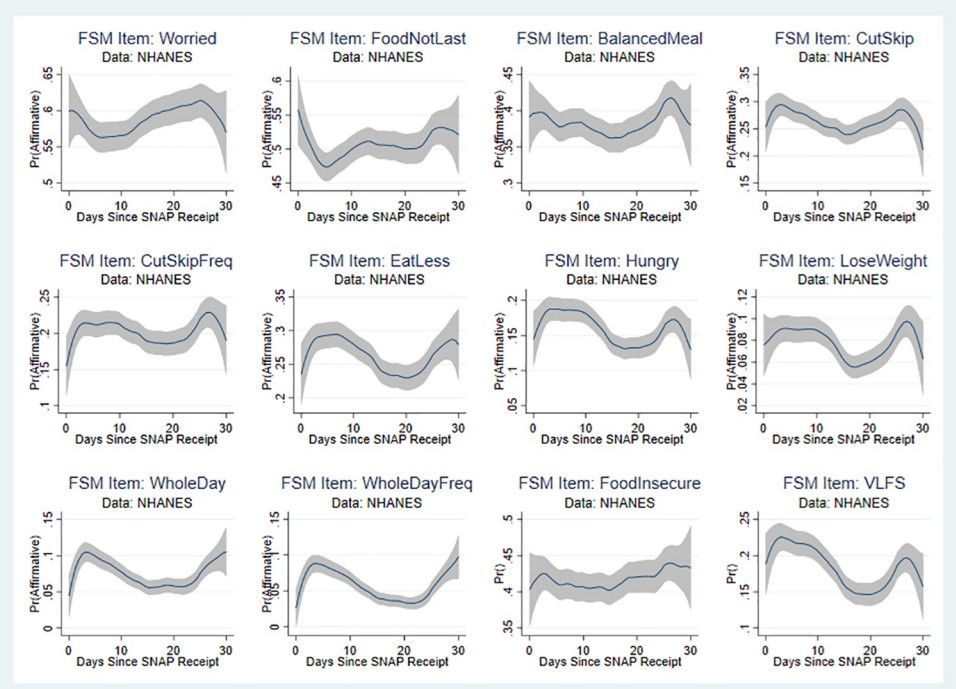

**Fig 1. Non-parametric regressions: Days since SNAP and food security items.** Source: Authors' calculations using NHANES data. N = 7,433. Regression estimates are weighted.

low food insecurity. (The rule-of-thumb bandwidth for these regressions was between 4 and 5; for consistency's sake we chose to use a bandwidth of 5 for all of them. We use an epanechnikov kernel. These choices do not affect the results.) The pattern of responses suggests an increase in the probability of affirmation for many of the items in the first week or ten days of the month and the last week of the month. Interestingly, there also seems to be a pronounced decrease in affirmations around the 20th of the month. This is especially evident for the more severe food security items: whether any adult ate less, was hungry, lost weight, or did not eat for a whole day. It is also evident for the very low food security indicator, although not for the food insecurity indicator (which includes low and very low food insecurity).

Figs 2 and 3 plot the marginal effects of being interviewed in a salience window on three FSM items and VLFS when we vary the salience window. (Figures like these are available for all of the items in the module upon request. We use these items for illustrative purposes.) Fig 2 shows the results for FSM items 6 and 7, which ask about whether anyone in the HH ate less or went hungry due to limited resources. Fig 3 shows the results for item 8, which asks whether any of the adults in the house lost weight because of lack of resources, and the binary indicator for VLFS. The results suggest that the uptick in affirmations near the end of the month begins to register as statistically significant when we include the last five or six days in the window. In addition, the window tends toward significant as we include at least the first week of the month along with the end of the month. These graphs don't show every combination of window choices, but they do show the window neighborhoods that appear to confirm what we observe in the non-parametric regressions. Namely, that there is a higher probability of affirmation for the first and last weeks of the month, and a pronounced decline in affirmation near the middle of the month.

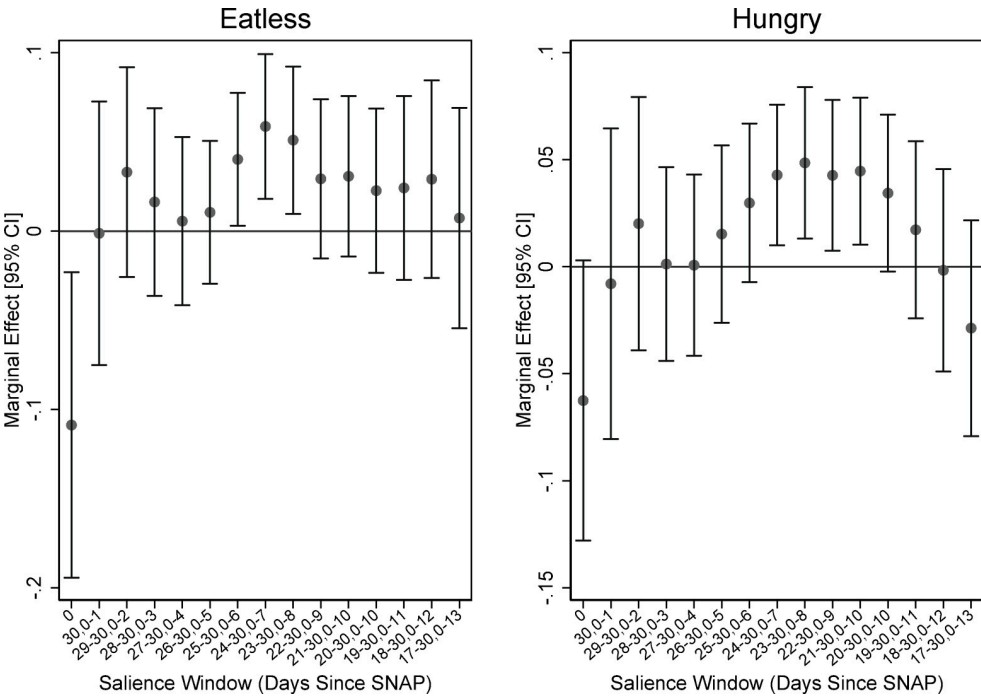

**Fig 2. Varying the bounds of salience window: FSM items eatless and hungry.** Source: Authors' calculations using NHANES data. N = 7,919. Regression estimates are weighted.

Guided by these results, we conduct the remainder of our analyses with a salience window that includes the first 7 and last 5 days of the SNAP month, although all of our results are robust to changes that include the first 10 and last 7 days of the month. The sample means of individual and household characteristics included in our regressions, stratified by whether they were within the salience window, are shown in Table 1. The two groups are very similar, except for some small differences in the education of the household reference person and in the proportion of that report at least $2,000 in liquid assets.

Table 2 shows the differences in the affirmations of the food security module items, food insecurity, and very low food security, once again stratified by whether or not the household was interviewed in the salience window. There are significant differences in the share of affirmations for the items that ask about cutting or skipping meals, eating less, being hungry, losing weight, and not eating for a whole day. Additionally, the total number of affirmations (the raw food insecurity score) is larger, and the probability of very low food security is higher, for those interviewed inside as opposed to outside of this window.

Our results going forward rely on the assumption that the "treatment"–being interviewed in the designated salience window–is randomly assigned across SNAP households. To test this, we regressed the variables in Table 1 on whether the household was in the treatment window: the p-value for an F-test of joint significance of the right hand side variables was 0.13. A model that used a probit specification had a p-value of 0.18. Thus, we feel comfortable moving forward assuming random assignment to the salience window.

## Regression on FSM items, raw score, food insecurity, VLFS

Table 3 shows the marginal effect of being interviewed in the salience window for each of the FSM items, the total number of affirmations, food insecurity, and very low food security

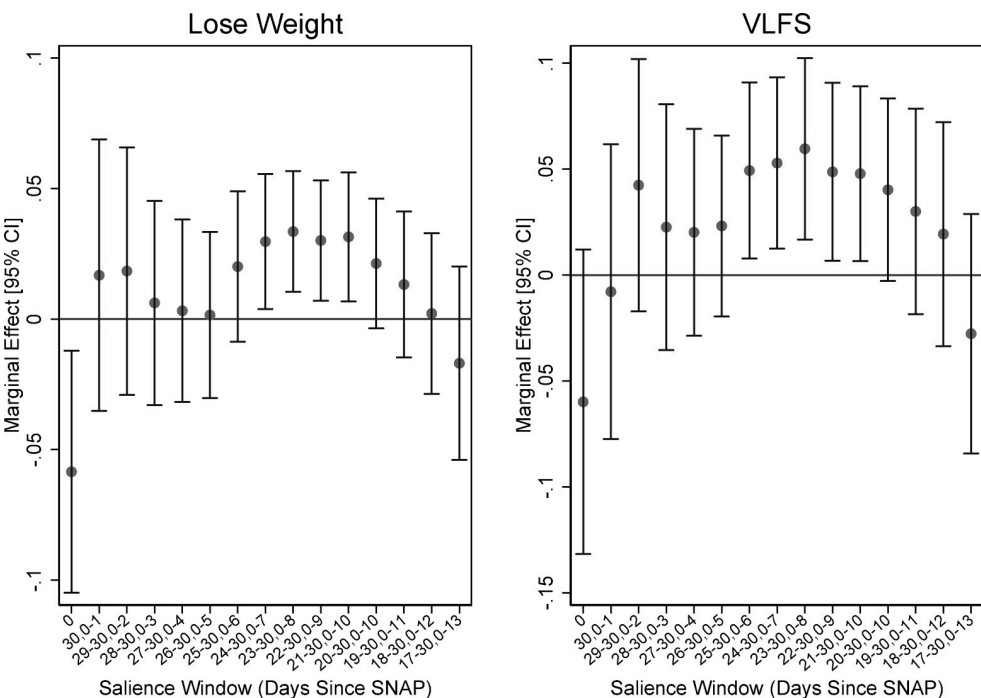

**Fig 3. Varying the bounds of salience window: FSM item lose weight and VLFS.** Source: Authors' calculations using NHANES data. N = 7,919. Regression estimates are weighted.

indicator variables. (Complete regression results are available in S1 Appendix.) The results indicate that affirmations of all of the items from item 4 (CutSkip) to the end of the FSM are affected by whether the family interview takes place in the salience window for SNAP benefits. The first three items, considered the least severe in the FSM, affirmations are not affected by when SNAP was last received, and neither is the indicator of food insecurity. The total number of affirmed FSM items and the probability of VLFS are significantly affected.

The results with respect to food security may seem puzzling because food insecure households are comprised of those with low and very low food security. An increase in VLFS would seem to imply an increase in food insecurity, since the latter is composed in part by the former. However, the estimate of the effect of being in the salience window on low food security is -0.03 (se = 0.022). The increase in the probability of food insecurity, while not significant, is a weighted average of this effect and that on VLFS.

## SNAP and 30-day FSM subsample

In order to gauge the degree to which the effect of SNAP is specific to 12-month food security questions, we examine the responses to individual 30-day food security items that were administered to a subsample of sample persons in the 2007–08 and 2009–10 waves of NHANES. (In separate analyses, we examined other consumption-related questions with 12-month or 30-day reference periods including frequency of alcohol consumption over 12 months, the number of drinks per day (conditional on drinking any) in the last 12 months, 30-day food spending, cigarettes smoked per day in the last 30 days, and food spending in the last 30 days. None of these measures showed the SNAP cycle related patterns that we observe with the food security items. Results available from the authors by request.) These persons belonged to households in which at least one of the first three food security items was affirmed on the

**Table 1. Sample means of independent variables.**

| | Not in Window | In Window | Difference |
|---|---|---|---|
| Male Sample Person | 0.453 | 0.434 | 0.019 |
| | (0.01) | (0.01) | (0.15) |
| Age of Sample Person | 28.008 | 27.893 | 0.114 |
| | (0.63) | (0.59) | (0.89) |
| Hispanic Sample Person | 0.455 | 0.467 | -0.011 |
| | (0.02) | (0.02) | (0.58) |
| Black Sample Person | 0.267 | 0.265 | 0.001 |
| | (0.03) | (0.03) | (0.94) |
| Other Race Sample Person | 0.253 | 0.257 | -0.004 |
| | (0.02) | (0.03) | (0.83) |
| Family Size | 0.064 | 0.057 | 0.007 |
| | (0.01) | (0.01) | (0.53) |
| HH Poverty-Income-Ratio | 3.915 | 4.077 | -0.162 |
| | (0.08) | (0.10) | (0.13) |
| HH Reference Person is Married | 0.901 | 0.857 | 0.044 |
| | (0.02) | (0.03) | (0.15) |
| HH Reference Person: No GED | 0.465 | 0.478 | -0.013 |
| | (0.02) | (0.02) | (0.57) |
| HH Reference Person: HS Grad | 0.39 | 0.41 | -0.01 |
| | (0.02) | (0.02) | (0.51) |
| HH Reference Person: Some College | 0.256 | 0.298 | -0.042* |
| | (0.02) | (0.02) | (0.05) |
| HH Reference Person: College Grad | 0.259 | 0.231 | 0.028 |
| | (0.01) | (0.01) | (0.15) |
| Have $2000 in Liquid Assets | 0.056 | 0.028 | 0.027** |
| | (0.01) | (0.01) | (0.03) |
| Participates in WIC | 0.035 | 0.032 | 0.003 |
| | (0.01) | (0.01) | (0.81) |
| N | 4,458 | 3,461 | |

Note

***$p < .01$

**$p < .05$

*$p < .01$ indicate that row differences are significant at indicated significance level. Source: Authors' calculations based on NHANES data. Summary statistics account for sample design and are weighted.

household module. These persons were asked to assess their individual food insecurity experience over the previous 30 days using the questions in Box 2. These questions were asked during the MEC interview, which generally occurs about 10 days after the initial household interview, so we adjusted the time since SNAP receipt to reflect the time before or after the family interview at which the MEC interview took place. (The NHANES public use file includes a variable indicating the number of days between the family and MEC interviews, DR1DBIH.)

We compare the individual 30-day responses to the household 12-month items in Figs 4–8. In order to make the 30-day cut and skip items comparable to the 12-month item, we summed the affirmations for 30-day cut and skip items to create 30-day rates of affirmation for cut/skip. These graphs indicate that the pattern of responses for the individual 30-day items differs

**Table 2. Sample means of dependent variables.**

| | Not In Window | In Window | Difference |
|---|---|---|---|
| FSM # 1, Worried | 0.585 | 0.591 | 0.006 |
| | (0.02) | (0.02) | (0.78) |
| FSM # 2, FoodNotLast | 0.503 | 0.506 | 0.003 |
| | (0.02) | (0.02) | (0.89) |
| FSM # 3, BalancedMeal | 0.370 | 0.399 | 0.029 |
| | (0.02) | (0.02) | (0.24) |
| FSM # 4, CutSkip | 0.244 | 0.296 | 0.052*** |
| | (0.01) | (0.02) | (0.01) |
| FSM # 5, CutSkipFreq | 0.191 | 0.222 | 0.030* |
| | (0.01) | (0.02) | (0.08) |
| FSM # 6, EatLess | 0.241 | 0.300 | 0.060*** |
| | (0.01) | (0.02) | (0.01) |
| FSM # 7, Hungry | 0.146 | 0.185 | 0.039** |
| | (0.01) | (0.02) | (0.04) |
| FSM # 8, LoseWeight | 0.070 | 0.094 | 0.024* |
| | (0.01) | (0.01) | (0.09) |
| FSM # 9, WholeDay | 0.057 | 0.104 | 0.047*** |
| | (0.01) | (0.01) | (0.00) |
| FSM # 10, WholeDayFreq | 0.041 | 0.085 | 0.044*** |
| | (0.01) | (0.01) | (0.00) |
| Raw Score (sum FSM 1–10) | 2.449 | 2.785 | 0.336*** |
| | (0.09) | (0.12) | (0.01) |
| Food Insecure | 0.404 | 0.433 | 0.029 |
| | (0.02) | (0.02) | (0.20) |
| Very Low Food Security | 0.165 | 0.219 | 0.053** |
| | (0.01) | (0.02) | (0.01) |
| N | 4,458 | 3,461 | |

Note

***p < .01

**p < .05

*p < .01 indicate that row differences are significant at indicated significance level. Source: Authors' calculations based on NHANES data. Summary statistics account for sample design and are weighted.

from the 12-month items. Instead of a distinct increase in affirmations during the first and last part of the month, with a sometimes sharp decline in affirmations around day 20 in the SNAP month, there is a more-or-less monotonic increase in affirmations across the month in the 30-day items.

The increase in affirmations of the individual 30-day FSM items at the end of the month is statistically significant for all but cut/skip (Table 4, right panel) and the monotonic increase is significant in the items capturing being hungry, losing weight, and not eating for a whole day (Table 4, left panel). (In results not shown, we explore whether there is also a cutpoint near the beginning of the SNAP month, and found that there was no significant break in the probability of affirmation.)

The evidence from Table 5 supports the idea that households are reporting consistently for the 30-day and 12-month items. The left panel shows the affirmations for the 30-day and 12-month items for households whose MEC interviews occurred in the last 6 days of the

**Table 3. Marginal effects of salience window on FSM items, raw score, food insecurity, VLFS.**

| Item | Marginal Effect | SE | Z |
|---|---|---|---|
| FSM # 1, Worried | 0.004 | (0.023) | 0.153 |
| FSM # 2, FoodNotLast | -0.001 | (0.020) | -0.044 |
| FSM # 3, BalancedMeal | 0.030 | (0.025) | 1.174 |
| FSM # 4, CutSkip | 0.055*** | (0.018) | 3.116 |
| FSM # 5, CutSkipFreq | 0.031* | (0.018) | 1.757 |
| FSM # 6, EatLess | 0.061*** | (0.020) | 3.013 |
| FSM # 7, Hungry | 0.044** | (0.019) | 2.379 |
| FSM # 8, LoseWeight | 0.027** | (0.013) | 2.031 |
| FSM # 9, WholeDay | 0.045*** | (0.015) | 2.935 |
| FSM # 10, WholeDayFreq | 0.042*** | (0.014) | 2.896 |
| RawScore | 0.339*** | (0.121) | 2.796 |
| Food Insecure | 0.027 | (0.022) | 1.212 |
| Very Low Food Security | 0.058*** | (0.021) | 2.807 |
| N | 7,919 | | |

Source: Authors' calculations using NHANES data. Regression estimates are weighted and take into account survey design information. Additional controls include all the variables listed in Table 1.

SNAP month and whose family interview and MEC interview occurred on the same day (left panel). This allows us to compare 30-day and 12-month FSM responses from the same interview date. The right panel compares the 30-day and 12-month affirmations for those whose MEC interview occurred 7 days or less after the family interview, providing a slightly larger sample and limiting variation in the reference points. For all of the items except cut/skip for persons whose family and MEC interviews were on the same day, the prevalence of 12-month items is higher than for corresponding 30-day items. (For the cut/skip comparison, the difference may be explained by the fact that the 30-day cut and skip questions were asked separately and we have treated them here as an indicator of whether either is affirmed, whereas the 12-month item is a single question.) In fact, for those households answering the 12-month food security and 30-day food security items on the same day, the rate of affirmation for the 12-month items is twice as for the 30-day items for LoseWeight and WholeDay (left panel). This is consistent with the idea that the 12-month FSM captures more food-insecure conditions given its longer reference period. We see similarly higher affirmations among the 12-month items among the larger sample all of individuals responding to the 30-day FSM within 7 days of the household response to the 12-month FSM.

The pattern of affirmations to the 30-day and 12-month FSM items are consistent with what is known about remembered utility: that summary reports tend to be a weighted sum of a peak or trough experience and the most recent period. The recalls at the end of the month would heavily weight the current decline in food spending and resources and intake. Affirmations at the beginning of the month would thus be a weighted average of the most recent period before SNAP receipt—the previous week (at the end of the previous month)—and, in the case of the 12-month FSM, a low point in the experience of food security over the year [17].

It is worth highlighting that the response patterns for the 30-day module reflect the increased salience of present experience in reporting on the previous 30 days. To see this, we can compare these responses to those of a recent small scale study of *daily* food insecurity across the SNAP month. Gassman-Pines and Schenk-Fontaine [27] administered food

*Box 2. Questions about individual food security experience in NHANES.* (Asked of persons in households that affirm at least one 12-month food security item during the household interview.)

1. In the last 30 days, did you cut the size of your meals because there wasn't enough money for food?

2. In the last 30 days, did you skip meal because there wasn't enough money for food?

3. In the last 30 days, did you eat less than you should because there wasn't enough money for food.

4. In the last 30 days, were you hungry but didn't eat because there wasn't enough money for food?

5. In the last 30 days, did you lose weight because you did not have enough money for food? (This question was not asked of sample persons less than 16 years old. For all other questions, an adult responded for all persons less than 12 years old. Adolescents (12–15) years old responded for themselves.)

6. In the last 30 days, did you not eat for a whole day because there wasn't enough money for food?

End of Box 2

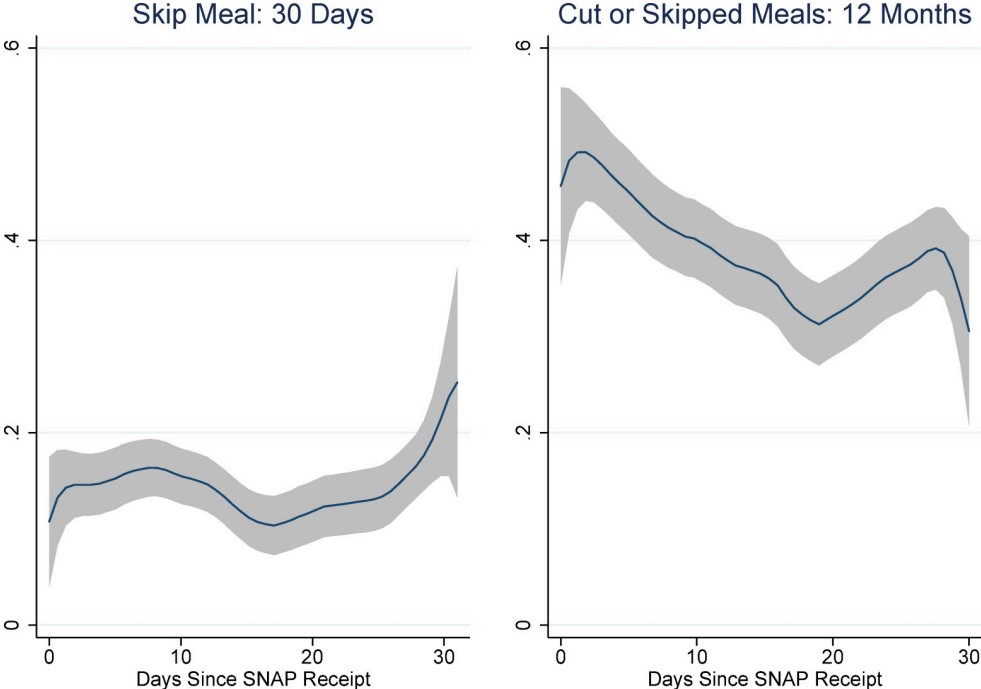

**Fig 4. Individual 30-day and household 12-month response to cut/skip FSM item and days since SNAP.** Source: Authors' calculations using NHANES data. N = 2,216. Non-parametric regression estimates are weighted.

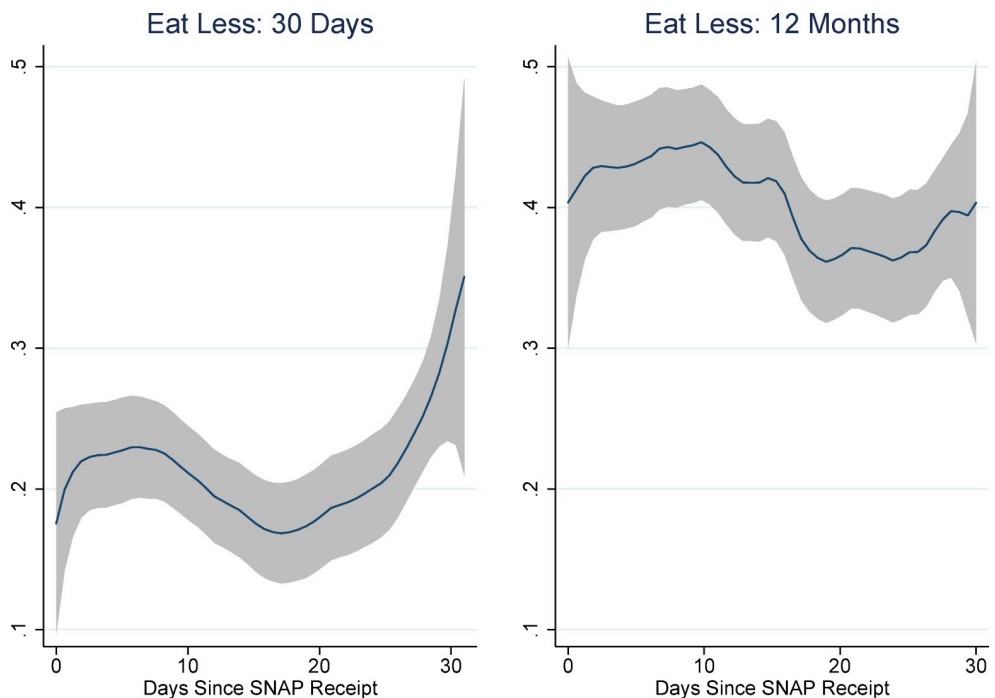

**Fig 5. Individual 30-day and household 12-month responses to eat less FSM item and days since SNAP.** Source: Authors' calculations using NHANES data. N = 2,218. Non-parametric regression estimates are weighted.

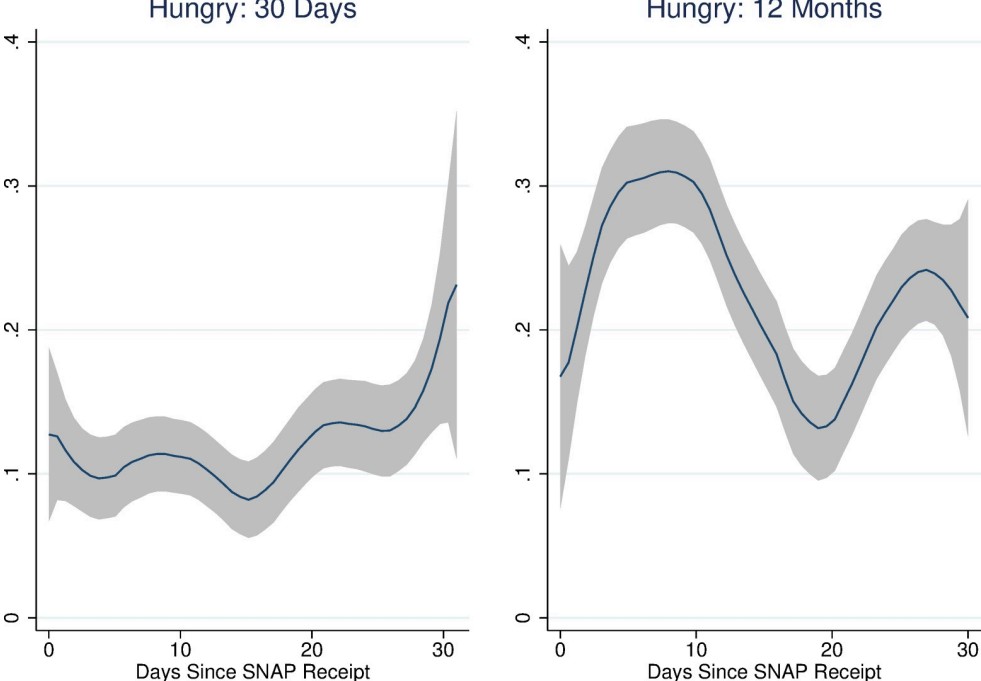

**Fig 6. Individual 30-day and household 12-month responses to hungry FSM item and days since SNAP.** Source: Authors' calculations using NHANES data. N = 2,216. Non-parametric regression estimates are weighted.

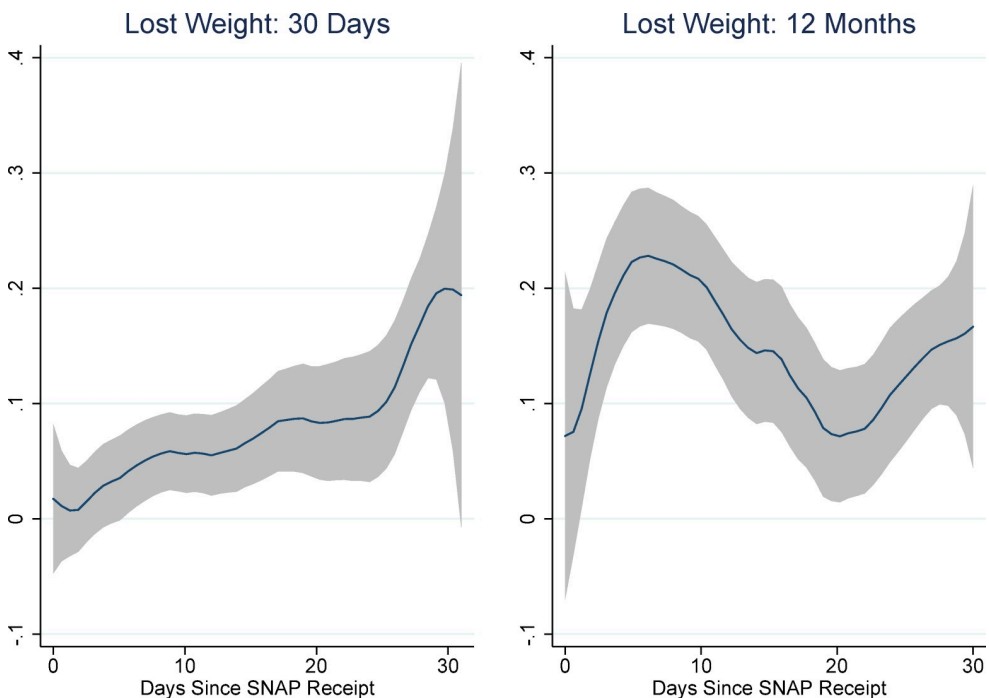

**Fig 7. Individual 30-day and household 12-month responses to lose weight FSM item and days since SNAP.**
Source: Authors' calculations using NHANES data. N = 647. Non-parametric regression estimates are weighted.

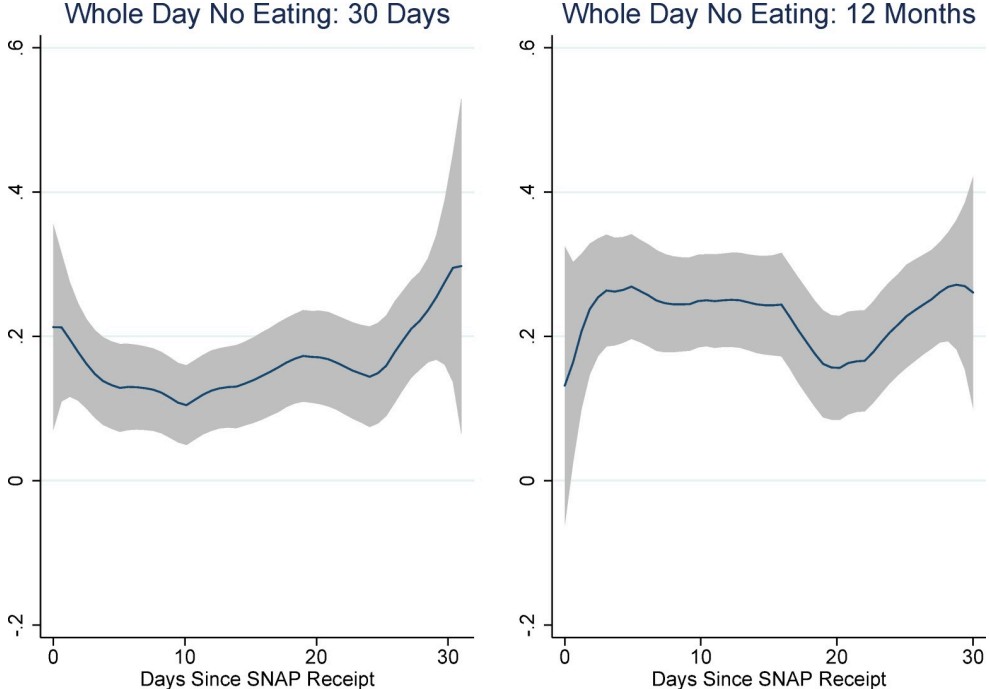

**Fig 8. Individual 30-day and household 12-month responses to not eat for a whole day FSM item and days since SNAP.** Source: Authors' calculations using NHANES data. N = 618. Non-parametric regression estimates are weighted.

**Table 4. Marginal effects of days since SNAP on 30-day food security items.**

| 30-Day Item | Days Since SNAP | | | Days Since SNAP > 24 | | | |
|---|---|---|---|---|---|---|---|
| | MFX | SE | Z | MFX | SE | Z | N |
| CutSkip | 0.001 | (0.001) | 0.639 | 0.032 | (0.030) | 1.096 | 2,216 |
| Eatless | 0.001 | (0.001) | 0.900 | 0.053** | (0.025) | 2.157 | 2,219 |
| Hungry | 0.002** | (0.001) | 2.217 | 0.037** | (0.019) | 1.985 | 2,218 |
| LoseWgt | 0.006*** | (0.001) | 5.168 | 0.120*** | (0.019) | 6.395 | 647 |
| WholeDay | 0.003* | (0.002) | 1.670 | 0.071** | (0.031) | 2.309 | 618 |
| N | 2,278 | | | | | | |

Source: Authors' calculations using NHANES data. Regressions are weighted and adjust for sample clustering and stratification.

hardship questions with daily reference periods to a sample of households in Durham, NC. They found a monotonic increase in the fraction of participants that affirmed food insecure conditions as the SNAP month went on (Fig 9), in very much the same way as the responses to the 30-day items in NHANES. Once again, this suggests that the 30-day estimated prevalence for individual food security items in NHANES highlight recent experiences of food hardship. They are likely biased downward by this foreshortened respondent recall.

A similar claim might be made about the 12-month recalls, although this case is complicated by the elevated affirmation rates at the beginning of the SNAP month. In what follows, we treat these affirmations as accurate recollections of past hardships, with the responses in the "middle" of the month indicative of under-reports of food hardships. While other assumptions are possible–we could assign a prior to the probability of over-reporting and use that to estimate the model–they don't matter for the percentage-point under-estimate of food hardships. We also point to recent work that gives statistical evidence of under- but not over-reporting of food hardships in the FSM [28].

## Bootstrap estimates of under-estimates of VLFS

To get a sense of how the recall bias in food hardships affects estimates of very low food security, we use a bootstrap procedure to draw from the distribution of the marginal effects shown in Table 3. For each bootstrap simulation, we first draw from the distribution of marginal effects for each question and simulate a new probability of affirmation based on that draw. We then sum the simulated affirmations over all items to get a simulated prevalence of very low food insecurity. When summing simulated affirmations, we obey the rules of internal

**Table 5. Prevalence of individual 30-Day and household 12-month food hardships at the end of SNAP month among households interviewed in the final 6 days of the SNAP month.**

| | MEC Interview: | | | | | |
|---|---|---|---|---|---|---|
| | Same Day as Family Interview | | | w/in 7 Days of Family Interview | | |
| | 12mo | 30day | N | 12mo | 30day | N |
| CutSkip | 0.381 | 0.476 | 21 | 0.311 | 0.289 | 135 |
| EatLess | 0.381 | 0.333 | 21 | 0.341 | 0.222 | 135 |
| Hungry | 0.333 | 0.190 | 21 | 0.261 | 0.104 | 134 |
| LoseWeight | 0.333 | 0.111 | 9 | 0.152 | 0.121 | 33 |
| WholeDay | 0.364 | 0.182 | 11 | 0.298 | 0.149 | 47 |

Source: Authors' calculations using NHANES data. Sample means are unweighted.

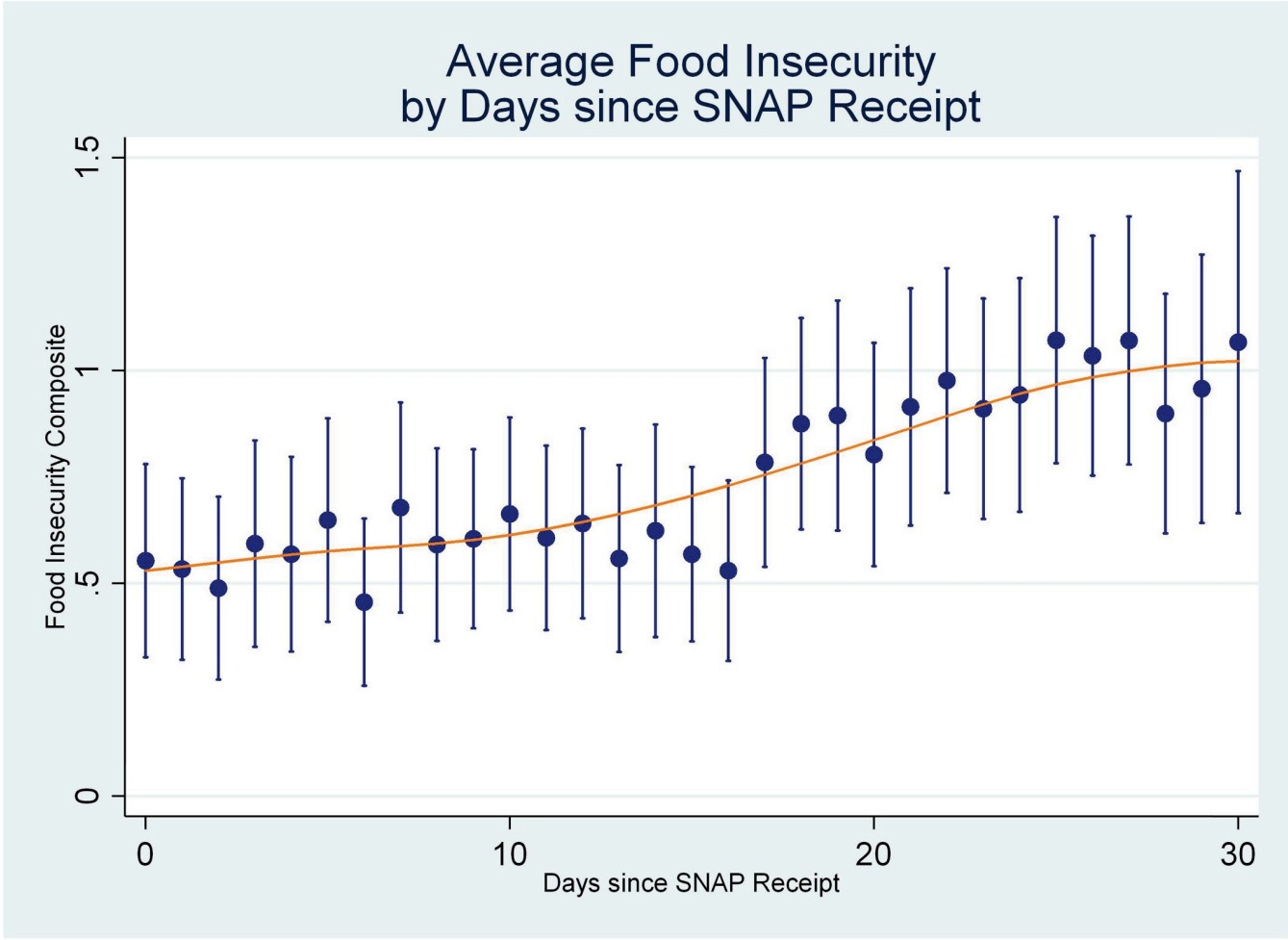

**Fig 9. Average daily food insecurity by days since SNAP receipt (Gassman-Pines & Schenck-Fontaine, 2019).** Source: Gassman-Pines, A. and Anika Schenck-Fontaine (2019, Fig 1). Reproduced by permission of the authors.

screening by which the food security module is administered: households without at least one simulated affirmation in the first three questions were not given the last 7 questions; households that did not affirm at least one of the items 4–8 were not given the last 2 items; and all questions not administered were counted as non-affirmations. As we mention above, we follow the procedure for bootstrapping in complex surveys outlined in Kolenikov [19] and Rao and Wu [20] using 300 bootstrap samples.

The results of this procedure are shown in Table 6. The first column is the marginal effect on VLFS reported in Table 3. Columns 2 and 3 show the underestimate in VLFS for SNAP recipients due to recall bias in percentage point terms (column 2) and relative to the prevalence for this group (percent, column 3). The next two columns show the same statistics for the low-income sample–those households at or below 200% of the FPL–and the last 2 columns show this for all households. The percentage point change in prevalence for the SNAP sample is about 3.2 points, roughly 17 percent of prevalence for this group; the prevalence reductions due to recall bias for the low-income sample and all households are roughly 9 and 8 percent, respectively.

**Table 6. Bootstrap estimates of under-estimate in very low food security prevalence.**

|  | (1) | (2) | (3) | (4) | (5) | (6) | (7) |
|---|---|---|---|---|---|---|---|
|  |  | SNAP households | | Low-income households | | All households | |
|  | MFX | Pct points | Prev (%) | Pct points | Prev (%) | Pct points | Prev (%) |
| Change | 0.058*** | 0.032*** | 0.169*** | 0.012*** | 0.088*** | 0.005*** | 0.078*** |
| Stderr | 0.021 | 0.012 | 0.062 | 0.004 | 0.033 | 0.002 | 0.029 |
| z | 2.807 | 2.716 | 2.716 | 2.679 | 2.679 | 2.664 | 2.664 |

Source: Authors' calculations using NHANES data. Bootstrap procedure accounts for survey stratification and clustering. Estimates use 300 bootstrap replications.

## Bounding the average treatment effect of SNAP on VLFS

Intuition suggests that downward bias in the prevalence estimate of VLFS would also have effects on estimates of the average treatment effect (ATE) of SNAP on VLFS. Instead of developing a point-identified treatment-effects model to estimate the potential bias in the average treatment effect (ATE), we estimate bounds on the ATE for those inside and outside the salience window. Our results make monotonicity assumptions consistent with the ideas that SNAP households are more likely to experience very low food security than non-SNAP households and that SNAP does not increase households' likelihood of VLFS. These assumptions are known as Monotone Treatment Selection (MTS) and Monotone Treatment Response (MTR), respectively [21, 22, 29]. We also vary errors in reporting of SNAP participation status, varying the share of misreports as well as whether errors are arbitrary or only under reports of participation occur (no false positives).

The results are shown in Table 7 and differentiate when households are surveyed in the salience window or not. The panels in the left 2 columns provide upper bounds on potential reductions in very low food security when we assume that reporting errors are arbitrary; the right panels shows the same results under the assumption that there are no false positives. Each of the rows details results for a different rate of classification error in SNAP participation. The rates shown are *potential* reductions in VLFS–that is, they have the "wrong" sign for treatment effects.

A characteristic of these estimates is that the potential reductions in very low food security are higher for households in the salience window compared to those outside it. For example, assuming 5 percent errors in classification of SNAP participation status, there is an increase in the potential reduction in VLFS from about 17 percent to 28 percent outside and in the window, respectively. Even larger differences are present when one assumes greater rates of

**Table 7. Bounding estimates of potential reductions in VLFS.**

| Upper Bounds on Potential Reductions in VLFS: MTR & MTS | | | | |
|---|---|---|---|---|
|  | Assumes Arbritrary Errors in reporting SNAP | | Assumes No False Positives in reporting SNAP | |
| SNAP Classification Error Rate | Not in Window | In Window | Not in Window | In Window |
| 0 | 0† | 0.068 | 0† | 0.068 |
| 0.05 | 0.17 | 0.279 | 0.168 | 0.279 |
| 0.1 | 0.181 | 0.448 | 0.174 | 0.445 |
| 0.25 | 0.226 | 0.973 | 0.174 | 0.445 |

† Lower bound for this estimate is -0.068. Lower bound for all other estimates is zero.

*Source*: Authors' calculations using NHANES data.

MTR = Monotone Treatment Response; MTS = Monotone Treatment Selection

misclassification of SNAP participation. Recent work by Meyer and Mittag [26] has shown that 25 percent misclassification would not be unusual if NHANES performed like other federal surveys. It is worth noting also that endogeneity is clearly a more serious problem than misreporting in the context of estimating ATE's, as has also been shown more generally when addressing misreporting [30]. The point here is that, even relying on the first moments of the data, methods to address misclassification of participation have more scope to find large treatment effects for households interviewed in the salience window than outside it.

## Discussion

This study shows that there are systematic differences in the affirmations rates of items in the household 12-month FSM across the SNAP month. In particular, we see higher levels of affirmations of more severe household FSM items and resulting prevalence of very low food security in the first 10 and last 6 days of the SNAP month. Although 30-day individual FSM items have elevated rates of affirmation in the last 6 days of the month, they lack the elevated rates at the beginning of the SNAP month. We assume that, for the 12-month household questions, the affirmations at the beginning month are accurate recollections prompted by SNAP receipt. With this in mind, we surmise that the lower response rates in the middle of the month are under-reports of food hardship and result in under-reporting of 12-month household VLFS among SNAP participants of 3.2 percentage points, or about 17 percent of prevalence. Similarly, we show that the ATE of SNAP on very low food security under these circumstances will also likely be biased downward.

There are a number of limitations of the current study. In particular, although Gregory and Smith (2019) found similar framing issues for the 30-day module administered in the National Food Acquisition and Purchase Survey (FoodAPS), their results pointed to biases in prevalence and ATE's of SNAP on food insecurity, not VLFS. We surmise that the longer reference period for the 12-month FSM items prompts a kind of recall that is related to more severe experiences of food hardship —in keeping with research about remembered utility —but we cannot test that hypothesis. We cannot examine 30-day *and* 12-month *household* food insecurity measures with this study, which might point the way to some possible causes for this difference. There are, in addition, differences in the overall framing in NHANES as opposed to FoodAPS that we cannot control for: among these is that the food security module is administered in the first interview for most NHANES participant households, but in the last interview for FoodAPS households. Another difference is that NHANES collected data on a wide range of health-related topics while FoodAPS focuses nearly entirely upon food acquisitions over a period of seven days. Once again, a single survey–like the Current Population Survey (CPS)–that administers both modules could help us get insight into these dynamics. (The current public version CPS does not have enough information to calculate days since SNAP receipt. Future research will access confidential data to look at 30-day and 12-month differences.)

Kahneman and Kruger (2006) recommend that surveys collecting subjective measures use the shortest reference period that is still useful. Interestingly, the results here and those of Gregory and Smith (2019) indicate that responses in *both* the 30-day and 12-month modules may be systematically influenced by SNAP receipt, although this influence is related to different measures of food insecurity. More particularly, this study gives evidence consistent with the idea that the FSM is subject to downward bias in estimates of prevalence; in addition to affecting our understanding of the who and how many experience food insecurity, this also biases our understanding of the efficacy of SNAP. A future study that examined both modules simultaneously could determine the differences in response patterns very specifically and perhaps improve the measurement of food insecurity.

## Supporting information

**S1 Appendix. Linear regression parameters.**
(DOCX)

## Author Contributions

**Conceptualization:** Christian A. Gregory, Jessica E. Todd.

**Data curation:** Christian A. Gregory, Jessica E. Todd.

**Formal analysis:** Christian A. Gregory, Jessica E. Todd.

**Investigation:** Christian A. Gregory.

**Methodology:** Christian A. Gregory, Jessica E. Todd.

**Project administration:** Christian A. Gregory.

**Software:** Christian A. Gregory.

**Validation:** Christian A. Gregory.

**Writing – original draft:** Christian A. Gregory, Jessica E. Todd.

**Writing – review & editing:** Christian A. Gregory, Jessica E. Todd.

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
