## [Decision Letter · Decision Letter 0]

7 Dec 2020

PONE-D-20-32057

SNAP Timing and Food Insecurity

PLOS ONE

Dear Dr. Gregory,

Thank you for submitting your manuscript to PLOS ONE. After careful consideration, we feel that it has merit but does not fully meet PLOS ONE’s publication criteria as it currently stands. Therefore, we invite you to submit a revised version of the manuscript that addresses the points raised during the review process.

Both reviewers like the paper and make some good suggestions. I also like the paper, and I think that it makes a significant contribution. I agree with the reviewers' comments. Most comments seem straightforward to address. The trickiest one is that the paper is hard to understand for someone unfamiliar with the literature. My suggestion is to re-write the introduction (the only section that most readers of the manuscript will actually read). It looks long for the paper's size, and you do not get to your contribution until the fourth page.   

We look forward to receiving your revised manuscript.

Kind regards,

Gabriel A. Picone

Academic Editor

PLOS ONE

Journal Requirements:

2. Please include your tables as part of your main manuscript and remove the individual files. Please note that supplementary tables (should remain/ be uploaded) as separate "supporting information" files

Reviewers' comments:

Reviewer's Responses to Questions

**Comments to the Author**

1. Is the manuscript technically sound, and do the data support the conclusions?

Reviewer #1: Yes

Reviewer #2: Yes

2. Has the statistical analysis been performed appropriately and rigorously? 

Reviewer #1: Yes

Reviewer #2: Yes

3. Have the authors made all data underlying the findings in their manuscript fully available?

Reviewer #1: Yes

Reviewer #2: Yes

4. Is the manuscript presented in an intelligible fashion and written in standard English?

Reviewer #1: Yes

Reviewer #2: Yes

5. Review Comments to the Author

Reviewer #1: PONE-D-20-32057_review

SNAP Timing and Food Insecurity

There has been some really cool research done using the variation in SNAP distribution dates within and across states within the last 5 years or so. SNAP receipt unlike other benefits is not perfectly aligned with timing of other income and is distributed over one to almost 20 days within a month depending on the state, according to a random case number assigned or using the last name of the recipient etc. Everyone receives it once a month but at different times within a calendar month according to the schedule adopted by the state, they live in.

The variation in timing and changes in the level of food security over the course of the benefit month has been studied as a mechanism in many of the above-mentioned work (though I am not giving any citations here, many of them are mentioned in this manuscript) and it is nice to see a paper exploring this mechanism as the outcome of interest. Though the paper is not about SNAP and food insecurity; It is an attempt to better understand the extend of food insecurity. By utilizing response difference in two sets of food insecurity questions (a 30-day recall versus a 12-month recall questionnaire) and differences in receipt timing relative to the interview date they try to gauge the extend of possible underestimation of food insecurity and additional underestimation of SNAP’s effectiveness in reducing this food insecurity due to salience of food insecurity, some behavioral reasons such like framing.

First of all, I think it is an interesting concept to study. Their way of trying to get at reporting differences as a function of salience of the benefit receipt is also a worthy endeavor. I think their econometric work is very solid.

My biggest problem is that there are too many things going on in the paper. It is hard to understand even for someone like me who is familiar with the literature. They need to do a better job stating their question/goal and maybe trim some of the analysis to an online supplement.

Here are my other issues:

Unlike other papers using SNAP timing they do not utilize the differences across states’ distribution schedules. Which can actually provide a nice dimension since in some states it is more uniformly distributed over the month than the others. It will be also nice to compare states where the SNAP month overlaps with the calendar month or other payment schedules and states where it is not.

The group for which they have data for 30-day and 12-month questions are cohorts from 2007-2008 and 2009 -2010. Can this be the source of the differences in what they see? These cohorts may have marginal new receivers of food stamps (there were many new entrants due to Great Recession) who may be experiencing food insecurity very differently. They may actually be not on food stamps over the course of the last year and had experience food insecurity unlike anything before and may have very different recall of the initial shock compared to their recent food insecurity experiences. Not sure if this is the case as I did not have time to think this through but if they have the data it may be really good to see if people who are recent SNAP recipients experience food insecurity SNAP schedule differently than long time receivers.

Salience window x-axis not sure what they capture in Figure 3 compared to other figures salience window axis. Do they have the same interpretation as the axis in later figures with salience windows, defined in days?

It is interesting that the 12-month recall correlates with the snap cycle. Is 30-day FS survey always after the 12-month survey?

Also, NHANES has some consumption data or some data on what they ate recently /nutrients in their body (I am not very familiar with this data set)? If this is indeed the case can we use this as a way to get it objective or subjective (framing) food insecurity? It may be interesting to look at nutrient deficiencies of 30-day vs 12-month VLFS people. Both within the group that has both surveys and for all 12-month respondents to address my earlier issue. This may be off target though and if so this again shows the paper is hard to follow at times.

Page 5 (second from the last line) should it be “food insecurity” and not “food security”

There are some squares in the text (looks like they are replacing long dashes)

Reviewer #2: November 28, 2020

Review of MS "SNAP Timing and Food Insecurity" (PONE-D-20-32057)

This paper uses data from NHANES to examine the effect of SNAP benefit receipt timing on 30-day and 12-month food insecurity, finding that timing of benefits vis-à-vis proximity to the survey matters. The consequence is that estimates of food insecurity, especially VLFS, are understated, and likewise that the effect of SNAP on VLFS is likewise understated.

This is an excellent paper, and important extension of the earlier work by one of the authors with Smith on SNAP salience. I think the paper is very close to publication, and I just have a few minor editorial suggestions, and one analysis suggestion.

1) On p. 9 when describing the data, you argue in favor of using the 10 adult focused questions to provide a common base of comparison across all households. I buy this, but it really begs for a set of analyses that are separated by whether there are children in the household. The authors are well aware of the literature that suggests that parents try to protect children from VLFS, or at least report that they do. It would thus be very useful if they conducted some of the baseline analyses on SNAP timing for households with kids and those without.

2) As I was reading the paper I kept asking myself “why aren’t they using the CPS?” You address this later in the paper during the Discussion, but I think it would be helpful if you brought that forward in the paper in the data section to help justify why you use the NHANES.

3) On p. 4 when you discuss literature on spending cycles, you should add references to Kuhn 2018 J Public Economics, and when you discuss geographic prices you should add a reference to Bronchetti, Christensen, and Hoynes 2019 J Health Economics

6. PLOS authors have the option to publish the peer review history of their article (what does this mean?). If published, this will include your full peer review and any attached files.

Reviewer #1: No

Reviewer #2: No

---

## [Author Response · Author response to Decision Letter 0]

7 Jan 2021

My biggest problem is that there are too many things going on in the paper. It is hard to understand even for someone like me who is familiar with the literature. They need to do a better job stating their question/goal and maybe trim some of the analysis to an online supplement. 

Response: We agree with this comment. We have trimmed all of the analysis of non-food-security items from the discussion of the paper, and summarized them in a footnote. Additionally, we have revised the introduction so that a minimum of material precedes the statement of our contribution – which now occurs in paragraph 3 of MS.

Here are my other issues:

Unlike other papers using SNAP timing they do not utilize the differences across states’ distribution schedules. Which can actually provide a nice dimension since in some states it is more uniformly distributed over the month than the others. It will be also nice to compare states where the SNAP month overlaps with the calendar month or other payment schedules and states where it is not.

Response: Our method relies on identification from knowing the actual date of SNAP receipt, rather than a range of possible dates – as would be the case if used state of residence to identify SNAP timing. In addition to the advantage of identification strength, this strategy has the practical advantage of not needing confidential restricted-access data, which looking at states would. The question about overlapping payment cycles for other programs has been examined in one of the references that we cite: Beatty et al. (2014) found that other payment cycles did not have any effect on the strength of the SNAP payment cycle. Additionally, Gregory and Smith (2019) looked at whether calendar- month-cycles (which overlap with payment cycles for things like SSI and SSDI) have a similar character to SNAP cycle; they found that calendar-month cycles were indistinguishable from noise. 

The group for which they have data for 30-day and 12-month questions are cohorts from 2007-2008 and 2009 -2010. Can this be the source of the differences in what they see? These cohorts may have marginal new receivers of food stamps (there were many new entrants due to Great Recession) who may be experiencing food insecurity very differently. They may actually be not on food stamps over the course of the last year and had experience food insecurity unlike anything before and may have very different recall of the initial shock compared to their recent food insecurity experiences. Not sure if this is the case as I did not have time to think this through but if they have the data it may be really good to see if people who are recent SNAP recipients experience food insecurity SNAP schedule differently than long time receivers.

Response: This is a really good point. One of the ways that we checked to make sure that new SNAP entrants in these years were not interpreting the items differently (or SNAP receipt differently) was to look at figures 4-8 to see if they resembled the full sample figures – and they do. Another thing that we do is to include wave dummies in all of our regressions to control for differences like this. It is really difficult to do more than this, since the data in NHANES don’t really allow us to characterize the SNAP spell length – we only know through looking at the demographics of SNAP during the recession that SNAP participants are generally whiter, in better health, with more education – on average a little more advantaged than in other waves. We have controlled for these observable characteristics as best we can in our regressions. 

Salience window x-axis not sure what they capture in Figure 3 compared to other figures salience window axis. Do they have the same interpretation as the axis in later figures with salience windows, defined in days? 

Response: In figures 2 and 3, the x-axis captures different lengths of salience window – these figures represent an attempt to capture a good range to use as the treatment variable in our regressions and subsequent analysis. In figures 4-8, the axis represents the number of days since SNAP receipt – this x-axis is essentially the same one used in all of the subfigures in figure 1. We changed the design of the X-axis so that it is easier to read and clarifies that the measure is the days since SNAP. 

It is interesting that the 12-month recall correlates with the snap cycle. Is 30-day FS survey always after the 12-month survey?

Response: One of our main points for this study was to look at the difference in response patterns for the 12-month (captured here) and 30-day modules (captured, for example in Gregory and Smith (2019)). In NHANES, the 30 day module was administered only in the two waves that we mention, and only for individuals in households that were at least marginally food insecure. Since it is administered (for these waves) in the MEC interview, it is after the 12-month FSM for most households in the NHANES sample.

Also, NHANES has some consumption data or some data on what they ate recently /nutrients in their body (I am not very familiar with this data set)? If this is indeed the case can we use this as a way to get it objective or subjective (framing) food insecurity? It may be interesting to look at nutrient deficiencies of 30-day vs 12-month VLFS people. Both within the group that has both surveys and for all 12-month respondents to address my earlier issue. This may be off target though and if so this again shows the paper is hard to follow at times. 

Response: Previous work, particularly Todd (2015), and Todd and Gregory (2018) examine how the SNAP cycle affects intakes – and find that it does vary in regular, predictable ways over the SNAP cycle. However, using the 30-day or 12-month VLFS sample to look at nutrients faces three problems. First, VLFS is a relatively rare event, so sample sizes are very small. Second, respondents who say that they have had VLFS over the last 30 days or 12 months may not show this on the intake diary, which covers one day usually after the 12-month FSM and one day before the 30-day FSM. Third, all but one of the items in the FSM addresses quantities rather than quality of food.

Page 5 (second from the last line) should it be “food insecurity” and not “food security”

There are some squares in the text (looks like they are replacing long dashes)

Response: Thank you for these. We have fixed these.

Reviewer 2

Reviewer #2: November 28, 2020

Review of MS "SNAP Timing and Food Insecurity" (PONE-D-20-32057)

This paper uses data from NHANES to examine the effect of SNAP benefit receipt timing on 30-day and 12-month food insecurity, finding that timing of benefits vis-à-vis proximity to the survey matters. The consequence is that estimates of food insecurity, especially VLFS, are understated, and likewise that the effect of SNAP on VLFS is likewise understated.

This is an excellent paper, and important extension of the earlier work by one of the authors with Smith on SNAP salience. I think the paper is very close to publication, and I just have a few minor editorial suggestions, and one analysis suggestion.

1) On p. 9 when describing the data, you argue in favor of using the 10 adult focused questions to provide a common base of comparison across all households. I buy this, but it really begs for a set of analyses that are separated by whether there are children in the household. The authors are well aware of the literature that suggests that parents try to protect children from VLFS, or at least report that they do. It would thus be very useful if they conducted some of the baseline analyses on SNAP timing for households with kids and those without.

Response: This is an important point that we considered in drafting the paper. Below is a table of the marginal effects of being in the salience window on each of the HH FSM items, food insecurity, and VLFS for households with children. These results are not meaningfully different from those for the full sample. Other specifications that we have run yield the same result. Further exploration of this is certainly needed, but it really demands its own study.

MFX of Salience Window on FSM Items: HH w Children

 MFX Se z

Worried -0.000 0.029 -0.007

FoodNotLast -0.003 0.024 -0.130

BalancedMeal 0.052 0.028 1.868

CutSkip 0.063 0.025 2.530

CutSkipFreq 0.044 0.023 1.915

EatLess 0.076 0.024 3.125

Hungry 0.047 0.023 2.023

LoseWeight 0.029 0.017 1.743

WholeDay 0.062 0.021 2.969

WholeDayFreq 0.057 0.020 2.886

FoodIns 0.039 0.025 1.541

VLFS 0.069 0.028 2.480

N 6,069 

2) As I was reading the paper I kept asking myself “why aren’t they using the CPS?” You address this later in the paper during the Discussion, but I think it would be helpful if you brought that forward in the paper in the data section to help justify why you use the NHANES.

Response: The public version of the CPS doesn’t contain the time since SNAP receipt. Although states are identified, this is not enough to identify the framing effects of this paper. Currently, we are working on getting access to confidential Census data to explore the question that you raise and that we outline in the Discussion section. We have added a sentence to the first paragraph of the data section to point this out.

3) On p. 4 when you discuss literature on spending cycles, you should add references to Kuhn 2018 J Public Economics, and when you discuss geographic prices you should add a reference to Bronchetti, Christensen, and Hoynes 2019 J Health Economics.

Response: We did cite the Kuhn paper, but the Bronchetti et al., although interesting, doesn’t really address cyclicality in SNAP distribution, so much as geographic differences in food prices. In reviewing recent work, we also added the reference to Carr and Packham (2019).

---

## [Decision Letter · Decision Letter 1]

29 Jan 2021

SNAP Timing and Food Insecurity

PONE-D-20-32057R1

Dear Dr. Gregory,

We’re pleased to inform you that your manuscript has been judged scientifically suitable for publication and will be formally accepted for publication once it meets all outstanding technical requirements.

One of the reviewers would like for you to add an additional reference. Within one week, you’ll receive an e-mail detailing the required amendments. When these have been addressed, you’ll receive a formal acceptance letter and your manuscript will be scheduled for publication.

Kind regards,

Gabriel A. Picone

Academic Editor

PLOS ONE

Additional Editor Comments (optional):

Reviewers' comments:

Reviewer's Responses to Questions

**Comments to the Author**

1. If the authors have adequately addressed your comments raised in a previous round of review and you feel that this manuscript is now acceptable for publication, you may indicate that here to bypass the “Comments to the Author” section, enter your conflict of interest statement in the “Confidential to Editor” section, and submit your "Accept" recommendation.

Reviewer #1: All comments have been addressed

Reviewer #2: All comments have been addressed

2. Is the manuscript technically sound, and do the data support the conclusions?

Reviewer #1: Yes

Reviewer #2: Yes

3. Has the statistical analysis been performed appropriately and rigorously? 

Reviewer #1: Yes

Reviewer #2: Yes

4. Have the authors made all data underlying the findings in their manuscript fully available?

Reviewer #1: Yes

Reviewer #2: Yes

5. Is the manuscript presented in an intelligible fashion and written in standard English?

Reviewer #1: Yes

Reviewer #2: Yes

6. Review Comments to the Author

Reviewer #1: Thank you for your revision. Manuscript reads much better now.

You added Carr and Packham (a paper I really like) and can consider adding one more reference as it is also about SNAP timing and maybe more directly related to food security related outcomes (through the mechanisms they discuss).

Cotti CD, Gordanier JM, Ozturk OD. Hunger pains? SNAP timing and emergency room visits. Journal of Health Economics. May 2020;71:102313.

The author show that children are relatively sheltered and prime aged adults potentially with children and older adults have more significant changes in their health care utilization. You can add it in the same paragraph (maybe even same sentence as CP paper)

thank you for you diligent work on an interesting topic.

Reviewer #2: (No Response)

7. PLOS authors have the option to publish the peer review history of their article (what does this mean?). If published, this will include your full peer review and any attached files.

Reviewer #1: No

Reviewer #2: No

---

## [Editor Report · Acceptance letter]

15 Feb 2021

PONE-D-20-32057R1 

SNAP timing and food insecurity 

Dear Dr. Gregory:

I'm pleased to inform you that your manuscript has been deemed suitable for publication in PLOS ONE. Congratulations! Your manuscript is now with our production department. 

Kind regards, 

on behalf of

Dr. Gabriel A. Picone 

Academic Editor

PLOS ONE